# Bioinspired light-driven chloride pump with helical porphyrin channels

Chao Li [1,2,4], Yi Zhai [1,4], Heming Jiang[3,4], Siqi Li [1], Pengxiang Liu [1], Longcheng Gao [1] ✉ & Lei Jiang [1,2]

Halorhodopsin, a light-driven chloride pump, utilizes photonic energy to drive chloride ions across biological membranes, regulating the ion balance and conveying biological information. In the light-driven chloride pump process, the chloride-binding chromophore (protonated Schiff base) is crucial, able to form the active center by absorbing light and triggering the transport cycle. Inspired by halorhodopsin, we demonstrate an artificial light-driven chloride pump using a helical porphyrin channel array with excellent photoactivity and specific chloride selectivity. The helical porphyrin channels are formed by a porphyrin-core star block copolymer, and the defects along the channels can be effectively repaired by doping a small number of porphyrins. The well-repaired porphyrin channel exhibits the light-driven Cl⁻ migration against a 3-fold concentration gradient, showing the ion pumping behavior. The bio-inspired artificial light-driven chloride pump provides a prospect for designing bioinspired responsive ion channel systems and high-performance optogenetics.

Membrane-embedded protein ion pumps and channels control the physiological activities by regulating the transport of ions across the cell membrane[1–3]. One of the most featured performances of the responsive ion channel is the membrane potential produced by the directional ion migration[4,5]. Halorhodopsin (HR) from the archaeon Halobacterium salinarum is a light-gated chloride channel, which functions as the sensory photo-receptor by light-driven chloride ion transport[6–8]. The insight into the HR structure reveals that the Schiff base group aggregates into the inner wall of the ion channel by the fold of protein, providing photo-receptor and selective chloride ion transport sites toward other anions such as F⁻, Br⁻, $H_2PO_4^-$, and $HCO_3^-$ (Fig. 1a)[9,10]. Deeping the understanding of HR structural characteristics and molecular mechanisms could inspire the development of extensive smart materials ranging from bioinspired ion channel systems for energy conversion[11–13] to the design of high-performance optogenetics[14–16].

Up to now, none of the artificial ion channels have achieved light-driven chloride ion pumps like the HR channel. Learning from the structure and mechanism of the HR channel, the artificial light-driven chloride pump should at least meet two basic requirements: the photo-receptor and the specific chloride selective site. Titanium dioxide ($TiO_2$)[17,18], porphyrin[19,20], spiropyran[21,22], graphitic carbon nitride[23], and azobenzene[24,25] are the most common materials used to construct the light-responsive channels for the effect of photo-induced electro or photo-isomerization. Due to the lack of chloride selective site, none of these channels exhibit the specific chloride selectivity as the HR channel does. To mimic the photo-driven chloride ion pump, light-responsive groups with specific chloride ion transport sites should be introduced into the nanochannel. Derived from the Schiff base structure of the HR channel, the porphyrin with the nitrogen-containing conjugated structure is supposed to be an ideal structure to achieve light-responsive chloride ion transport. Aligning the porphyrin into the

[1]Laboratory of Bio-Inspired Smart Interfacial Science and Technology of Ministry of Education, School of Chemistry, Beihang University, Beijing 100191, P. R. China. [2]Key Laboratory of Bio-inspired Materials and Interfacial Science, Technical Institute of Physics and Chemistry, Chinese Academy of Sciences, Beijing 100190, P. R. China. [3]Shenzhen Bay Laboratory, Shenzhen 518132, China. [4]These authors contributed equally: Chao Li, Yi Zhai, Heming Jiang. ✉ e-mail: lcgao@buaa.edu.cn

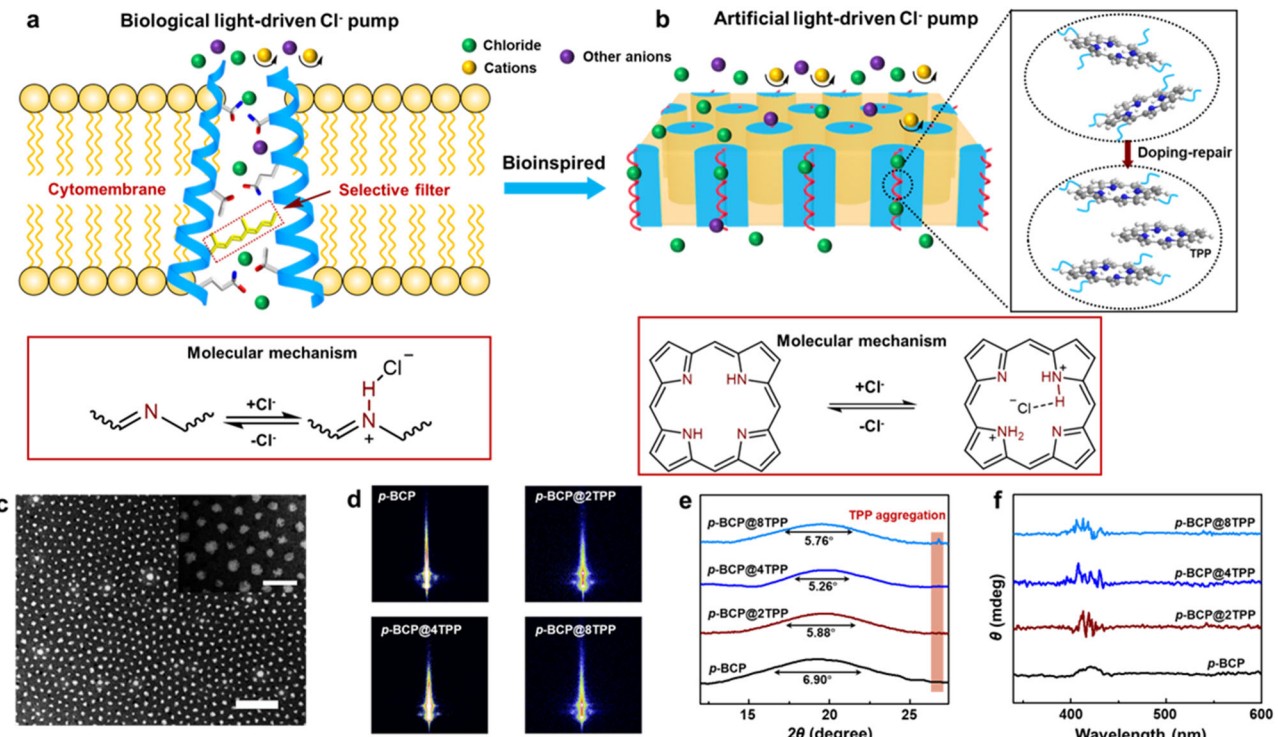

**Fig. 1 | Bioinspired light-driven chloride channel. a** An illustrated diagram of the biological HR channel with Schiff base group serving as the selective filter and photo-receptor. **b** Schematic of bioinspired light-driven chloride pump with porphyrin acting as the photo-receptor and selective ion transport site. The defects in the helical channel are repaired by the doping of TPP. **c** TEM image of *p*-BCP@4TPP, exhibiting periodic nanocylinder structure (Scale bar: 500 nm). Inset: an enlarged TEM image (Scale bar: 100 nm). **d** 2D GI-SAXS patterns of *p*-BCP@*n*TPP membranes, indicating vertically aligned nanocylinder structure. **e** 1D intensity profiles of corresponding 2D WAXD patterns, indicating porphyrin doping-repaired aggregates. **f** CD spectra of *p*-BCP@*n*TPP membranes. The positive signals at 417 nm indicate the helical porphyrin channel structure for membranes.

channel and utilizing the cavity for chloride transport could achieve the light-driven chloride ion pump.

Here, we apply the fabrication strategy of biological HR channels to align a small content of porphyrin into the helical channels under the synergetic interaction, exhibiting a high diffusion current density of 0.68 mA·cm$^{-2}$ with the light-driven chloride selective transport (Fig. 1b). The helical porphyrin channel is self-assembled by a porphyrin-cored block copolymer (*p*-BCP) and quantitively doping its porphyrin initiator (TPP) to repair the inevitable defects along the channels. As a result, the porphyrin *d*-spacing decreases, resulting in the decrement of ion transport resistance. The helical channel exhibits excellent Cl$^-$ selectivity toward cations and anions like F$^-$, Br$^-$, H$_2$PO$_4^-$, and HCO$_3^-$. Under the driven force with the light radiation, the Cl$^-$ could undergo the directional migration even without the external potential or concentration gradient, converting solar energy into electronic energy. The power density of the light-driven ion pump reaches 56.0 mW·m$^{-2}$ under the symmetric concentration gradient. Furthermore, Cl$^-$ could migrate against a 3-fold concentration gradient under light irradiation. The design strategy for the light-responsive channel with the least functional groups and the mechanism of light-driven Cl$^-$ transport could highlight the prospect in the fields of the fabrication of bioinspired smart systems and the high-efficiency harvesting for solar energy.

## Results

### Fabrication of doping-repaired helical porphyrin channels

The synthesis of *p*-BCP was described in Supplementary Information. The chemical characterizations are shown in Supplementary Figs. 1–4. As each *p*-BCP contains only one porphyrin, the mass content of porphyrin is less than 1%. Despite such low contents, porphyrin aggregates into a high-density helical porphyrin channel array under

the synergetic interaction of BCP self-assembly and porphyrin π-π stacking (Fig. 1c, d). However, due to the steric hindrance of the connected polymer chains and the chain entanglement, porphyrin stacking is inevitably frustrated, causing defects in the porphyrin channels. That is the main reason for the big full width at half-maximum (FWHM) at the scattering peak on the WAXD curve (Fig. 1e). The stacking frustration could reduce the ion conductivity of the porphyrin channels. To mitigate stacking frustration, free porphyrins are introduced into the BCP system, which acts as a repairing agent[26], and bridges the adjacent porphyrin core in *p*-BCP[27,28]. Different TPP contents are doped into the *p*-BCP system, defined as *p*-BCP@*n*TPP, where *n* is the TPP doping number per *p*-BCP (Supplementary Fig. 5). Considering the high molecular weight, the total porphyrin content is still very low. As shown in Fig. 1d and Supplementary Fig. 6, the results of GI-SAXS patterns and SAXS profiles remain almost unchanged, which shows the TPP doping does not change the periodic nanocylinder structure. The periodic length is calculated to be ~70.6 nm, which is well consistent with the TEM image. From the cross-section SEM image (Supplementary Fig. 7), we can see transmembrane nanocylinder structure. At the sub-nanometer scale, porphyrins are confined in the nanocylinder and stacked through π-π interaction, which conformed by the wide-angle X-ray diffraction (WAXD) spectra. With TPP doping, the *d*-spacing and FWHM of porphyrin aggregation slightly decreased, indicating doping-induced defect repairing. The *d*-spacing reaches its minimum value when the doping number is 4 (Fig. 1e, Supplementary Fig. 8, and Supplementary Table 1). Interestingly, when the doping number increases to 8, a slight increment of *d*-spacing is seen. It indicates that the excess TPP tends to form aggregation instead of repairing the porphyrin channels, as a diffraction signal (2θ of 26.8°) related to TPP aggregation is detected (Supplementary Fig. 9). Furthermore, a remarkable N$_{1s}$ signal is observed for *p*-BCP@8TPP on the

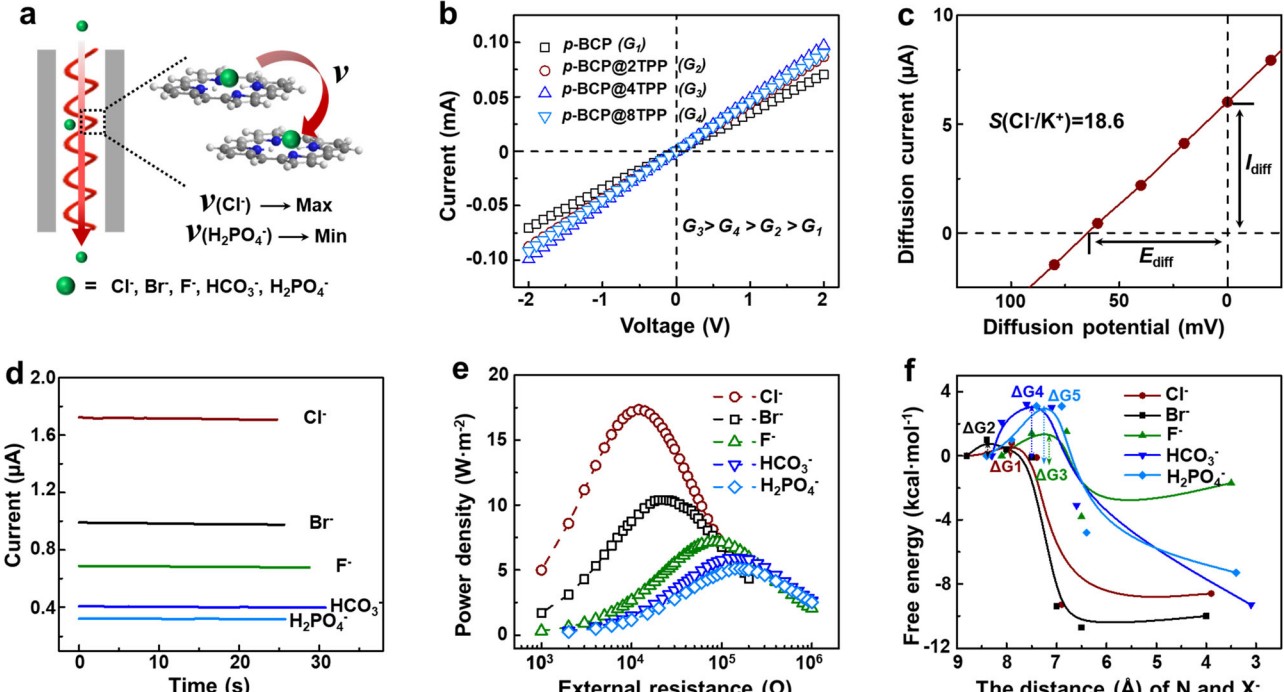

**Fig. 2 | The specific Cl⁻ selectivity with helical porphyrin channels. a** An illustrated diagram of ion transport through the helical porphyrin channel. The transport rate of Cl⁻ is the fastest. **b** The current-voltage curves of the $p$-BCP@$n$TPP samples. $p$-BCP@4TPP exhibits the highest conductance with the appropriate TPP doping. **c** The current-voltage curve of the $p$-BCP@4TPP membrane under a 0.5 M/0.01 M concentration gradient, indicating an excellent anion selectivity toward cations. **d** Current-time curves of the $p$-BCP@4TPP membrane with different electrolytes, showing the highest ion transport current for Cl⁻. **e** The power densities of the $p$-BCP@4TPP membrane as a function of the external resistance with different anionic electrolytes (0.5 M/0.01 M). KCl electrolyte exhibits the highest power density. **f** PES section of anion transformation between adjacent porphyrins along the shortening of X⁻-N distance coordinate, X⁻ representing Cl⁻, Br⁻, F⁻, HCO₃⁻, and H₂PO₄⁻.

X-ray photoelectron spectroscopy (XPS), confirming the tiny TPP-aggregation (Supplementary Figs. 10–12). The porphyrins, including the ones on BCPs and the doped ones, are aligned into a high density of helical porphyrin channel, which can be proved by the circular dichroism (CD) tests. As shown in Fig. 1f, all the samples show positive Cotton effect signals at 417 nm. The helical channels result from the porphyrin J-aggregation. From the UV-vis spectra, we can see weak absorption peaks at 728 nm in the Q-band region, corresponding to J-type aggregation (Supplementary Fig. 13). The same conclusion can also be made from the fluorescence spectra, where the peak at 717 nm is attributed to J-aggregation (Supplementary Fig. 14).

## Specific chloride ion selectivity

The porphyrin channels provide specific pathways for Cl⁻ toward cations and other anions (Fig. 2a). From the current-voltage curves under symmetric solutions, we can see that all the porphyrin channels show good ionic conductance, as the $p$-BCP@4TPP shows the highest conductivity (Fig. 2b). The excellent Cl⁻ conductivity is theoretically investigated by density functional theory (DFT) calculations. To simplify the simulation, we choose two adjacent porphyrins as the model to simulate the transport process and calculate the relative free energy barrier for Cl⁻ transport. For the $p$-BCP membrane, the relative free energy barrier for Cl⁻ migration is the highest, indicating the highest ionic resistance. While for the $p$-BCP@4TPP membrane, the relative free energy barrier for Cl⁻ migration is the lowest, indicating the lowest ionic resistance (Supplementary Fig. 15 and Supplementary Data 1), which is consistent with the experimental results of Fig. 2b.

Compared to K⁺, Cl⁻ shows stronger affinity with porphyrin. As a result, the porphyrin channel shows Cl⁻ selectivity towards K⁺ (Supplementary Fig. 16). The Cl⁻ selectivity is experimentally researched by the I-V curve under an asymmetric concentration

gradient (0.5 M/10 µM). The ion migration contribution to the ionic current from the low-concentration side is negligible. As shown in Supplementary Fig. 17, the ionic current at the positive voltage (Cl⁻) is prominently higher than the ionic current at the negative voltage (K⁺), indicating much higher Cl⁻ transport efficiency. The Cl⁻ selectivity is quantitatively investigated by the I–V curve under a 50-fold concentration gradient (Fig. 2c). By subtracting the redox potential and redox current with a salt bridge[29,30], the diffusion potential ($E_{diff}$) is directly obtained by the open voltage of the I-V curve, which is resulted from the Cl⁻ selective transport under the concentration gradient. The ion selectivity ratio ($S$) is defined as the ratio of $D_{Cl^-}/D_{K^+}$, where $D_{Cl^-}$ and $D_{K^+}$ mean the diffusion coefficients for Cl⁻ and K⁺, respectively. The diffusion potential is related to the selectivity of the channel via the Goldman-Hodgkin-Katz voltage equation[31,32]:

$$E = \frac{k_b T}{e} \ln\left(\frac{S a_H + a_L}{S a_L + a_H}\right) \tag{1}$$

Where $E$, $k_b$, $T$, and $e$ represent the diffusion potential, Boltzmann constant, absolute temperature, and electron charge, respectively; and $a_H$ and $a_L$ are the activities of Cl⁻ for the high- and low-concentration solutions, respectively. The Cl⁻ selectivity would be calculated to be ~18.6, indicating a much faster Cl⁻ migration compared to that of K⁺ migration in the helical porphyrin channel. The Cl⁻ selectivity is also confirmed by the simulation results, as the relative free energy barrier for Cl⁻ migration is much lower than those of cations (Supplementary Fig. 18).

The selectivity of Cl⁻ toward anions is researched by conductance comparison in different electrolytes, whose cations are all set as K⁺. From the I–V curves, we can see that the conductance in the KCl

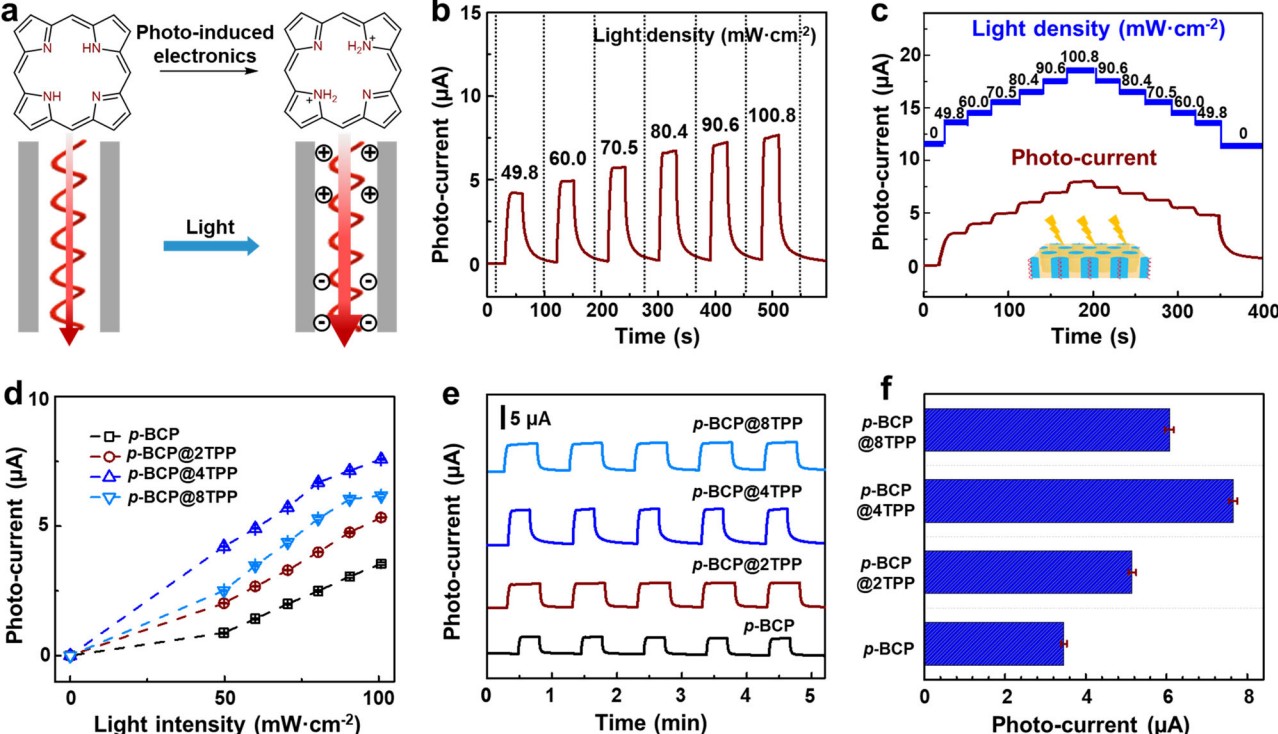

**Fig. 3 | Light-responsive ion transport with helical porphyrin channels. a** An illustrated diagram of light-responsive ion transport. The light-induced surface charges facilitate the Cl⁻ transport. **b** The time-dependent photocurrent curve of the *p*-BCP@4TPP membrane under a +1.0 V bias with different light densities, exhibiting increased photocurrent value with the increased light density. **c** The time-dependent photocurrent curve of the *p*-BCP@4TPP membrane, revealing excellent reversibility. **d** The photocurrent of the *p*-BCP@*n*TPP membranes as a function of different light densities. **e** The time-dependent curves of the *p*-BCP@*n*TPP membranes at a light density of 100.8 mW·cm⁻². **f** The comparison of photocurrent for the *p*-BCP@*n*TPP membranes at a light intensity of 100.8 mW·cm⁻².

solution is the highest, while the KH₂PO₄ solution shows the lowest (Supplementary Fig. 19). We also directly compare the conductance in different electrolytes under an external 1.0 V bias. As shown in Fig. 2d, the current obtained from the KCl electrolyte is much higher than those of other electrolytes. The Cl⁻ selectivity, defined as the current ratio between Cl⁻ and other anions, are similar to the biological HR channel (Supplementary Fig. 20)[7]. The Cl⁻ selectivity arises from the reversible porphyrin-Cl⁻ complex through hydrogen bonds (Supplementary Fig. 21)[33,34].

The differences in anion selectivity could also be reflected by the osmotic energy conversion ability with different electrolytes, which is also reflected in previous literature[35]. The *p*-BCP@4TPP membrane exhibits the highest power density of 17.4 W·m⁻² with a KCl electrolyte (under 50 concentration gradient, Fig. 2e and Supplementary Fig. 22). Compared to previous work, the ultrahigh power density of helical porphyrin membrane contributes to three aspects: the high-density helical porphyrin channel, the continuous and short ion transport pathway, as well as swelling suppression[36,37]. The Cl⁻ selectivity toward other anions is theoretically studied. For the transformation process, we calculated potential energy surface (PES) sections of different ion transformations between adjacent porphyrins along the shortening of anion-nitrogen distance coordinate (Fig. 2f). The PES sections of all anions exhibit similar relative free energy variation trends, indicating they go through a similar transformation process. While different anions showed different relative free energy barriers (ΔG), indicating the difficulty of their transformation process is different. For the above five ions, ΔG increases by the order of Cl⁻, Br⁻, F⁻, HCO₃⁻ and H₂PO₄⁻ (Supplementary Fig. 23), indicating the Cl⁻ migration is the easiest to occur and the H₂PO₄⁻ migration is the most difficult to occur, which are consistent with experimental results.

## Light-responsive ion transport

Owing to the photochemical property of the porphyrin, the helical channels exhibit a light-responsive ion transport with features of high sensitivity, fast response speed, and high reversibility (Fig. 3a). The electrochemical equipment for light-responsive ion transport is schematized in Supplementary Fig. 24, and the membranes are mounted evaluate the merit of light-responsive behaviors. As shown in Fig. 3b, the time-dependent photocurrent curves were measured under a +1.0 V with different light intensities. The wavelength is set as 405 ± 5 nm, according to the maximum absorption in the UV-vis spectra. To simplify the analysis, the net current caused by light irradiation is calculated as $I_p = I_{light} - I_{dark}$. The helical porphyrin channel membranes show fast and reversible responses toward light radiation. As the light intensity increases, accompanied by a higher surface charge density and faster ion transport, the photocurrent increases. It is worth noting that the Cl⁻ selectivity is enhanced upon light radiation (Supplementary Fig. 25). The reversibility of photo-response is further explored by the photocurrent curve under a cycle of step-increased light density and step-decreased light density (Fig. 3c). The excellent reversibility of the photo-responsive property is resulted from the stability of the membrane structure. The porphyrin channel membrane shows structural stability toward light irradiation as the nano-cylinder nanostructure and the porphyrin aggregation remains almost unchanged after the light stimulation (Supplementary Figs. 26, 27). The content of the doped TPP also influences light-responsive ion transport. As shown in Fig. 3d and Supplementary Fig. 28, the photo-currents of all the samples experience an almost linear increase in light intensity. All membranes demonstrate excellent reversibility, which is evidenced by the five-cycle photocurrent curve with excellent stability and signal-to-noise ratio (Fig. 3e and Supplementary Fig. 29). The photocurrent of the membranes at a light intensity of 100.8 mW·cm⁻²

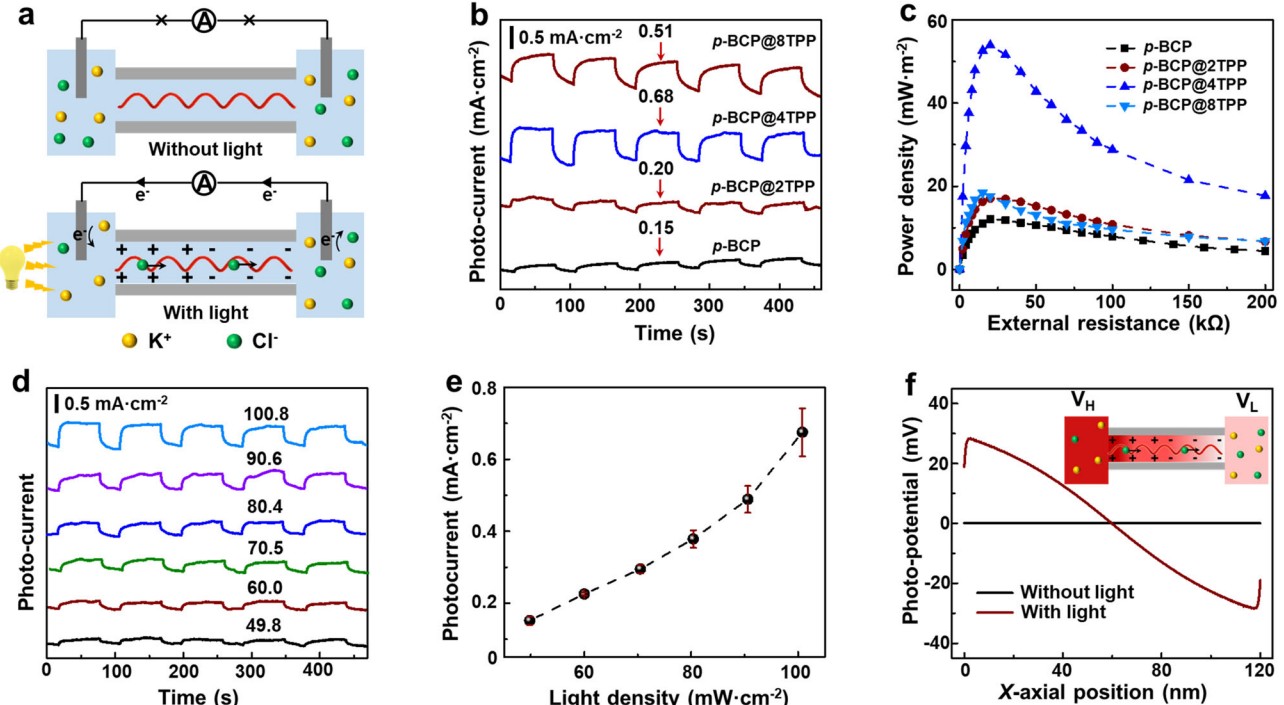

**Fig. 4 | Photoelectric energy conversion with helical porphyrin channels. a** An illustrated diagram of the light-driven ion pump. The directional Cl⁻ migration induces the diffusion current under a light-driven force. **b** The time-dependent photocurrent densities of the *p*-BCP@*n*TPP membranes under the symmetric electrolyte, showing the highest current density for the *p*-BCP@4TPP membrane. **c** The power densities of the *p*-BCP@*n*TPP membranes as a function of the external resistance under the symmetric concentration electrolyte. **d** The time-dependent

photocurrent densities of the *p*-BCP@4TPP membrane under different light densities. **e** The photocurrent densities of the *p*-BCP@4TPP membrane as a function of light densities, exhibiting higher photocurrent density for higher light density. **f** The calculated electrical potential profile along the *X*-axis of a single porphyrin channel with and without light irradiation. Error bars represent standard deviation ($n = 3$).

is summarized in Fig. 3f. The *p*-BCP@4TPP membrane exhibits the highest photocurrent.

## Photoelectric energy conversion

Like the HR channel, the helical porphyrin channel could achieve directional Cl⁻ transport under the light-drive force, converting the solar energy into electrochemical energy. The redistribution of surface charge is regarded as the dominant factor for the light-driven ion pump[38,39]. The light induced positive charge on surface is confirmed by the surface photovoltage spectroscopy test (Supplementary Fig. 30). Although the light irradiation might cause the increase of membrane temperature, the contribution can be negligible (Supplementary Fig. 31), because the content of porphyrins is low, while the light is weak, and meanwhile, the tiny heat is quickly conducted by the electrolyte. The mechanism of the light-driven ion pump is illustrated in Fig. 4a. Without light irradiation, there is no potential bias to drive Cl⁻ across the membrane in between symmetric electrolytes. Thus, no diffusion current is detected. With light irradiation, the electrons of porphyrin separate from the holes and move to the illuminated side, causing an asymmetric surface charge distribution. Subsequently, Cl⁻ is attracted and enriched on the positive membrane side, causing a higher Cl⁻ concentration. Meanwhile, Cl⁻ is repelled and dissipated on the other negative membrane, resulting in a lower Cl⁻ concentration. The Cl⁻ concentration gradient on the two sides of the membrane surface caused by the redistributed surface charge drives Cl⁻ to transport along the porphyrin channels, producing a diffusion current and diffusion potential. As shown in Supplementary Fig. 32, the light-induced current density experiences a dramatic increment with the concentration increasing from 1 mM to 0.1 M. This is because the low concentration electrolyte indicates the high ion transport resistance for the doped membrane. While with further concentration increasing,

the Debye length becomes shorter (0.3 nm for 1.0 M KCl)[40–42], lowering the Cl⁻ selectivity and decreasing the photocurrent. Thus, the optimal concentration for light-driven Cl⁻ selective transport is 0.1 M. The light-induced diffusion current and potential could be confirmed by the I–V curve in a symmetric electrolyte with and without light irradiation. The open circuit voltage ($U_{oc}$) and short current, obtained from the I-V curve, are the contribution of ion-selective migration under light irradiation (Supplementary Fig. 33). The light-induced current densities are researched by the time-dependent current curves in a 0.1 M symmetric electrolyte under 100.8 mW·cm⁻². The *p*-BCP@4TPP membrane shows the highest photocurrent densities of 0.68 mA·cm⁻² (Fig. 4b and Supplementary Fig. 34). Since the light-driven Cl⁻ selective transport under a symmetric concentration produces the photopotential, the membrane could be mounted into an electrochemical cell, functioning as the electric generator. Then, the electrochemical cell is connected with an external resistance to evaluate the maximum output power. The currents under different external resistances are recorded and the power density ($P$) could be calculated by the equation: $P = I^2 R/S$, where, $I$, $R$, and $S$ represent the recorded current, external resistance, and the membrane area, respectively. The power density of the light-induced power generator reaches its maximum value when the external resistance is equal to the internal resistance, indicating a 56.0 mW·m⁻² for the *p*-BCP@4TPP membrane (Fig. 4c and supplementary Fig. 35). The conversion efficiency of light to electricity is calculated to be ~0.27%. Furthermore, the light density directly determines the photocurrent density, as the more asymmetric surface charge distribution is constructed with the increased light density. Thus, we can see that the photocurrent density dramatically increases with the light density, ranging from 0.15 to 0.68 mA·cm⁻² (Fig. 4d, e). The light-induced diffusion potential is semi-quantitatively studied based on the Poisson and Nernst-Planck (PNP) equations. The

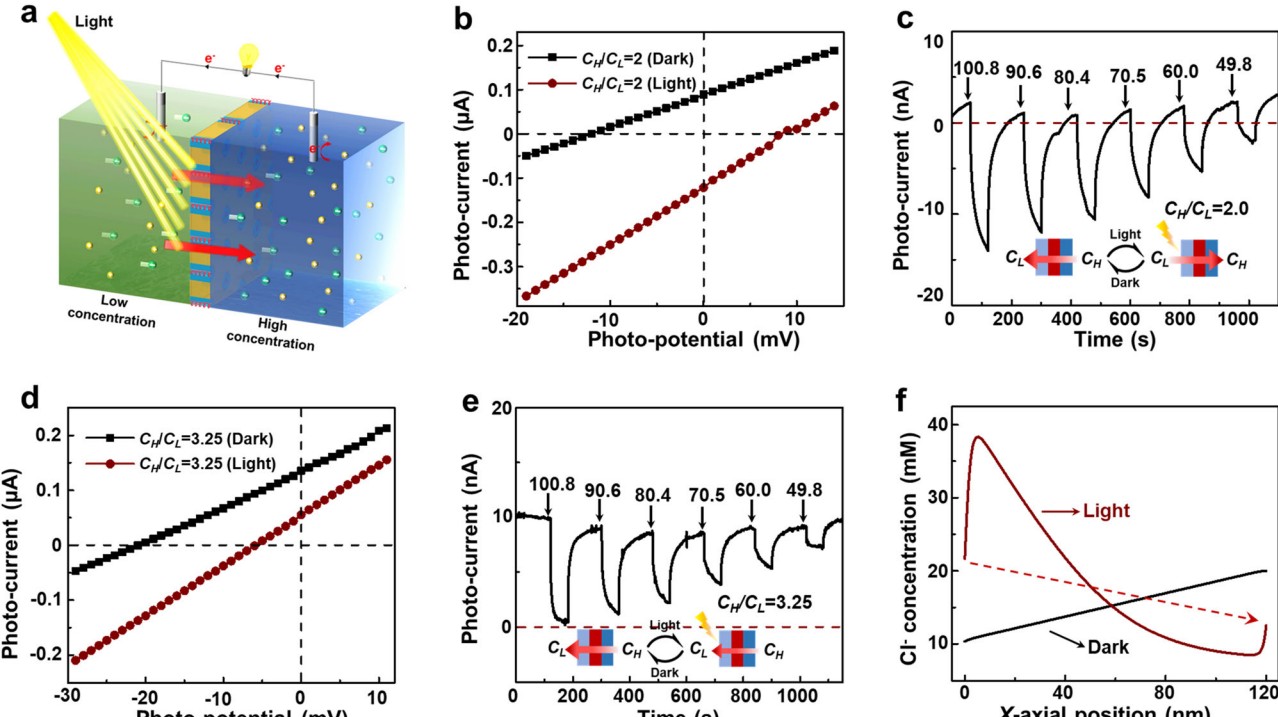

**Fig. 5 | Light-driven chloride ion pump with helical porphyrin channels. a** An illustrated diagram of the light-driven ion transport inversing the concentration gradient. The charge redistribution of the $p$-BCP@4TPP membrane surface provides the power for inversing-concentration-gradient ion transport. **b** Current-voltage curves of the $p$-BCP@4TPP membrane in an asymmetric electrolyte ($C_H$/$C_L$ = 2) with and without light irradiation (100.8 mW·cm$^{-2}$). **c** The time-dependent photocurrent of the $p$-BCP@4TPP membrane under an asymmetric concentration gradient ($C_H$/$C_L$ = 2) with different light densities. The inversed current indicates the

inversed ion transport directions. **d** Current-voltage curves of the $p$-BCP@4TPP membrane in an asymmetric electrolyte ($C_H$/$C_L$ = 3.25) with and without light irradiation (100.8 mW·cm$^{-2}$). **e** The time-dependent photocurrent of the $p$-BCP@4TPP membrane under an asymmetric concentration gradient ($C_H$/$C_L$ = 3.25) with different light densities. The decreased current indicates that light irradiation blocks the ion transport along the concentration gradient. **f** The calculated Cl$^-$ concentration distribution profiles along the $X$-axis of a single porphyrin channel under 2-fold and 3.25-fold concentration gradients with and without light irradiation.

calculated electrical potential profile along the axis of a single porphyrin channel is shown in Fig. 4f. Without light irradiation, a slightly higher potential is observed for the ion attraction in the nanochannel entrance. And with light irradiation, the asymmetric charge distribution directly causes a potential drop along the $X$-axial of a single channel, indicating a higher potential on the illuminated side and a lower potential on the unilluminated side. The theoretical results from the simulation are well in agreement with the electrochemical results. The bioinspired light-driven chloride pump provides insight into efficient solar energy conversion.

**Light-driven chloride ion pump**
Like the HR channel, the helical porphyrin channel also acts as a light-driven chloride pump. Photo energy could redistribute the surface charge, producing build-in electric-field that drive upgradient ion transport (Fig. 5a)[43–45]. Without light irradiation, the Cl$^-$ selectively migrates from the high-concentration side to the low-concentration side, forming the diffusion potential and diffusion current. When applying light irradiation at the low-concentration side, the internal electric field of the membrane propels migration of Cl$^-$ from low to high concentration, causing an inversed diffusion potential (negative) and diffusion current (positive) (Fig. 5b). The inversed ion migration with light irradiation is also observed by the time-dependent current curve ($C_H$/$C_L$ = 2). As seen in Fig. 5c, the current signal turns from positive to negative with light irradiation. And with the decreased light density, the photocurrent value becomes closer to the zero-current line, proving the lower light-induced diffusion current. Even under a 3-fold gradient, the built-in electric field can achieve Cl$^-$ pump (Supplementary Figs. 36, 37). When the gradient reaches 3.25-fold, the

force of the concentration gradient dominates the ion transport, as Cl$^-$ migrates from the high-concentration side to the low-concentration side (Fig. 5d, e). We further investigated the effect of pH value on the ion pump property. Under acidic conditions, the porphyrin is fully protonated and Cl$^-$ selectivity of the membrane increases (Supplementary Figs. 38, 39). The light-induced electric field could drive Cl$^-$ migration against a 2.0-fold concentration gradient (Supplementary Fig. 40). While under alkalic conditions, the membrane exhibits a lower Cl$^-$ selectivity and could achieve active Cl$^-$ migration against a 4.0-fold concentration gradient with light irradiation (Supplementary Fig. 41). The differences in ion pump properties arise from the differences in surface charge densities. The higher surface charge density (acidic condition) indicates the weaker redistributed surface charge density with light irradiation, resulting in a smaller light-driven force. The inversed-concentration-gradient ion diffusion phenomenon under a 2-fold concentration gradient is also semi-quantitatively researched by the simulation results. The simulation model is shown in Supplementary Fig. 42. The light irradiation redistributes the surface charge density of the membrane, resulting in the illustrated side being positively charged and the unillustrated side being negatively charged. As the difference in charge density between the two sides increases, the concentration gradient along the single porphyrin channel would reverse (Fig. 5f and Supplementary Fig. 43). The ion accumulation at the low-concentration side and the ion depletion at the high-concentration side could drive the Cl$^-$ migration against the concentration gradient. The concentration decrement at the low-concentration entrance and the concentration increment at the high-concentration entrance are contributed from the bulk solution.

## Discussion

Inspired by the structural features of the biological HR channel, we apply the biological atomic economic strategy to align the porphyrin into high-density helical channels under the synergetic interaction of BCP self-assembly and π-π stacking. To repair the defects in the helical channels, The TPP molecule is doped into the porphyrin channel, without sacrificing the nano-cylinder structure and short porphyrin distance. Thus, the repaired helical channels exhibit improved ionic conductance. Owing to the photoelectrochemical property of porphyrins, the helical channels exhibit light-responsive Cl⁻ transport with features of high stability, fast responsive speed, and excellent reversibility. Besides, the light-driven ion transport with excellent Cl⁻ selectivity could convert the solar energy into the diffusion current, reaching a value of $0.68\,mA\cdot cm^{-2}$ and generating a power density of $56.0\,mW\cdot cm^{-2}$. The light-driven ion diffusion is ascribed to the redistribution of the surface charge, producing a built-in electric field, which could also provide the driven force for the ion diffusion against the concentration-gradient direction. This bio-inspired light-driven chloride channel with an aggregated-group strategy could provide a prospect for designing complex bioinspired systems and highlight the prospect of converting solar energy into electric energy.

## Methods

**Materials.** Materials, synthesis of the sample, and the preparation of the doped membrane are described in the Supplementary Information.

### Characterization

$^{1}$H NMR measurements were performed with Bruker AV-500 spectrometer. Gel permeation chromatography was measured by a Waters 2410 instrument equipped with a Waters 2410 Ultraviolet detector. TEM images were recorded using JEM-2100. SEM image was recorded by ZEISS Gemini 300. GI-SAXS, 2D SAXS, and 2D WAXD measurements of membranes were carried out with a Xeuss 3.0 system (Xenocs SA, France). The SPV test was carried out with the CEL-SPS1000 instrument. Zeta potentials of prepared membranes were determined with Anton Paar SurPASS. The CD spectra were tested with the JASCO instrument (J-815). The UV-vis spectra were recorded with Shimadzu UV-3600. The fluorescence emission properties of samples were detected with a Nanolog FL3-2iHR fluorescence spectrometer. XPS spectra were measured by Thermo Scientific K-Alpha. The electrochemical measurements were performed by a Keithley 6487 semiconductor picoammeter at room temperature. The detailed testing conditions can be seen in the Supplementary Information.

### Numerical simulation

The simulated model for the porphyrin channel is described in the Supplementary Information. The numerical simulation was performed based on the Poisson and Nernst-Planck (PNP) equations.

The Nernst-Planck Eq. (2) that describes the transport properties of charged nanochannels is shown below:

$$j_i = D_i \left( \nabla c_i + \frac{z_i F c_i}{RT} \nabla \varphi \right) \qquad (2)$$

where, $j_i$, $D_i$, $c_i$, $z_i$, $F$, $R$, $T$, and $\varphi$ are the ionic flux, diffusion coefficient, ion concentration for each species, valence number for each species, Faraday constant, the universal gas constant, Kelvin temperature, and electrical potential, respectively.

The electrical potential inside the nanochannel can be related to the ionic concentration by the Poisson Eq. (3):

$$\nabla^2 \varphi = -\frac{F}{\varepsilon} \sum z_i c_i \qquad (3)$$

Simplified by assuming steady-state conditions, the flux should satisfy the time-independent continuity Eq. (4):

$$\nabla \cdot j_i = 0 \qquad (4)$$

The diffusion coefficients towards K⁺ and Cl⁻ are $1.96 \times 10^{-9}$ and $2.03 \times 10^{-9}\,m^2 \cdot s^{-1}$. The valence numbers for K⁺ and Cl⁻ are +1 and −1. The temperature is 298 K. The dielectric constant of the aqueous solution was assumed to be 80. For the simulation of ion transport regulation under light irradiation, the concentration gradient of the electrolyte is set as 2.0-fold.

## Data availability

All data generated in this study are provided in the article and Supplementary Information, and the raw data generated in this study are provided in the Source Data file. Source data are provided with this paper. These data are available from the authors on request. Source data are provided with this paper.

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

## Acknowledgements

This work is financially supported by the National Natural Science Foundation of China (22175009), the National Key Research and Development Program of China (2022YFB3805900), Postdoctoral Fellowship Program of CPSF (GZB20230923), and the Fundamental Research Funds for the Central Universities. We thank Dr. Y. Liu (Research Institute for Frontier Science) for assistance with the GI-SAXS and WAXD tests.

## Author contributions

L.J., L.G., and C.L. conceived the idea. C.L. designed the detailed project scope. C.L. and Y.Z. performed all experiments with the help of S.L. and P.L. C.L. and H.J. carried out the numerical simulations. All authors discussed the results and wrote the manuscript together.

## Competing interests

The authors declare no competing interests.
