## [Peer Review File · Nature Communications]

Bioinspired light-driven chloride pump with helical porphyrin channelsREVIEWER COMMENTS

Reviewer #1 (Remarks to the Author):

This manuscript reports an interesting light-enhanced chloride pump in a helical porphyrin channel array membrane. Characterizations are well conducted to show the molecular structure of the proposed helical porphyrin channels and results are solid to demonstrate the proposed idea. In general, this work is informative in the field and may inspire some new directions of light-responsive selective and power generation devices. However, before publication, there are still some concerns that should be addressed in the revision. After proper revision, I would be happy to support the manuscript for publication.

1. In Figures S2 and S3, the molecular structures should be clearly identified like Figure S1.
2. The zeta potentials of the membrane before and after light stimulation are required, to show its ion selectivity.
3. Figure 2g indicates that the developed membrane can output a very high power density of 17.4 W/m² under a 50-fold KCl gradient, compared with the previously reported values of 4-7 W/M² (e.g., Nano Energy 223, 105, 108007; Adv. Funct. Mater. 2023, 33, 2211316; Adv. Funct. Mater. 2022, 32, 2204068). Please explain why so high performance compared with existing works.
4. What is the testing area for measuring ion transport and osmotic power?
5. Figure 2e shows high Cl anion selectivity of the proposed membrane, compared with other anions, which is one of the main contributions of this work. I am curious how to measure in situ measure the current change when replacing the electrolyte. The detailed experimental measurement method should be mentioned. Another similar study of highly anion selective membrane (Sci. Adv. 2021, 7, eabe9924) should be credited.
6. I am curious why a fixed 2-fold or 3.25-fold concentration gradient is applied in Figure 5? Any physical meaning?
7. What is the structural stability (e.g., SEM, TEM, and x-ray data) of the proposed membrane after light stimulation for a while (e.g., half or one hour)?
8. Any data to support the π - π stacking?

Reviewer #2 (Remarks to the Author):

This manuscript reports the development of a bioinspired light-driven chloride pump. The membrane pump is made using star block blocks with porphyrins at their center. Additional porphyrins are doped into the membrane to help create continuous channels across the membrane. A variety of characterization and transport tests are utilized to demonstrate the structure and interesting transport characteristics of the resulting films. The manuscript may be of interest to the readership of your journal but significant work is needed to communicate the main findings of the manuscript in a clearer and more concise manner. The authors have clearly worked hard to develop a detailed study of the light-driven chloride pumps. However, the manuscript presents the results of this extensive investigation without

much detailed analysis or communicating clearly the hypothesis or motivation behind each experiment. I strongly suggest that the authors consider what information is absolutely essential for communicating their main findings and reduce the complexity of the figures. I think this will make assessing the underlying technological advances easier.

A small detail that is important. The authors reference the porphyrin content of the copolymer materials. However, within the main text, I don't believe they ever clarify if it is a mole, mass, or volume percent.

It is important for the authors to execute control experiments based around star copolymer membranes that do not contain the porphyrin.

The reported membranes are selective toward the transport of chloride. This appears to be true whether the membrane surface is illuminated or not. Does the selectivity of the membrane change at all when the surface is illuminated. The authors should consider executing experiments with a reverse bias. If the chloride was attempting to permeate through the negatively charged unilluminated side, transport should be slow due to electrostatic repulsion and an asymmetric I-V curve distinct from those shown in Figure S15 should arise. This experimental result would also help to support the authors' hypothesis regarding charge redistribution driving transport upon illumination.

The authors should discuss the energetic efficiency of the light-driven transport process.

I am not convinced that the numerical simulations provide useful insights since the results are controlled by the boundary conditions, which are set by the authors. The comparison that is made is really just manifesting change that was a direct result of the researchers' input not a phenomenon that emerges naturally from the interactions within the system.

The concentration profile in Figure 5f deserves further analysis and discussion. Why don't the concentrations at the boundary of the two cases align with each other? Additionally, the local curvature of the concentration profiles for the illuminated systems suggests that transport is being driven from right to left (similar to the dark cases). More details regarding these simulations are needed for others to be able to repeat them.

Reviewer #3 (Remarks to the Author):

This manuscript reports a photoactive helical porphyrin channel that shows specific chloride selectivity. It is demonstrated that this porphyrin channel can act like a chloride ion pump under light irradiation, which drives Cl⁻ migration against a 3-fold concentration gradient. Although a series of experiments have been conducted in this work, there are problems with the experimental results and explanations. Therefore, we do not think this work can be published at the present stage.

1. It is not necessary to emphasize "doping-repaired" in the title. The core of this work is the porphyrin-

channel based chloride ion pump. The whole system can also work with undoped helical porphyrin channel.

2. It is known that porphyrin molecules are generally negatively charged and easily coordinate with metal ions. Therefore, it is confusing that the porphyrin channel preferentially binds to anions and shows anion selectivity in this work.

3. What is the isoelectric point of a porphyrin molecule? The zeta potential of the porphyrin-core star block copolymer needs to be measured.

4. The influence of pH on the chloride selectivity and chloride ion pump should be given. It is a very important control experiment.

5. In Figure 1c, a clearer and enlarged TEM image is required. What is the thickness of the membrane? The SEM image of the cross section of the membrane should be given.

6. The salt bridge is not shown in Figure S17. The photo of the actual device used in this work also needs to be shown.

7. About the anion/cation selectivity of the membrane, a more intuitive and effective experiment is recommended, which can be referred to "Figure 4a" in a previous work (DOI: 10.1021/jacs.5b09918).

8. "Page 5 Line 136" It is described that "the Cl⁻ selectivity is contributed by the formation of halogen bonds". What is the molecular structure of this "halogen bond"? In addition, ref.33 is quoted here to explain this selectivity. However, the porphyrin molecules reported in Ref.33 are metal-ion centered, which is a totally different story. An explanation is needed here.

9. The light irradiation might also cause the increase of membrane temperature or even a temperature gradient inside the membrane. This will not only increase the ion transport rate, but also drive the ion transport along the temperature gradient. Therefore, the influence of temperature should also be discussed.

10. It is described that the ion pump is mainly caused by the light-induced surface charge redistribution. However, when the surface charge is redistributed, the whole system should reach a steady state without any further ion migration. What causes the continuous ion flow?

11. The redistribution of the surface charge under light irradiation may also lead to a reversal of ion selectivity. This ion selectivity reversion can also cause the reversion of the direction of current and membrane potential, showing a similar data reported in this work. The anion/cation selectivity of the membrane under light irradiation needs to be measured. Further discussion needs to be added in the manuscript.

12. Scale bar of current in Figure 3e and Figure 4b,d,g is missing.

Reply To Reviewers

Reviewer #1 (Remarks to the Author):

This manuscript reports an interesting light-enhanced chloride pump in a helical porphyrin channel array membrane. Characterizations are well conducted to show the molecular structure of the proposed helical porphyrin channels and results are solid to demonstrate the proposed idea. In general, this work is informative in the field and may inspire some new directions of light-responsive selective and power generation devices. However, before publication, there are still some concerns that should be addressed in the revision. After proper revision, I would be happy to support the manuscript for publication.

Reply: Thanks for your positive comments and helpful suggestions. We will respond to your comment point-in-point as follows.

Comment 1. In Figures S2 and S3, the molecular structures should be clearly identified like Figure S1.

Reply: Thanks for your helpful suggestions.

We have revised the Figures S2 and S3. The revised ^1H NMR spectra are shown in Figures R1 and R2.

Figure R1. ^1H NMR spectra of macroinitiator.

Figure R2. ^1H NMR spectra of *p*-BCP.

Comment 2. The zeta potentials of the membrane before and after light stimulation are required, to show its ion selectivity.

Reply: Thanks for your helpful suggestions.

Unfortunately, the instrument for measuring zeta potential does not support in-situ detection under light stimulation. We have taken another test (surface photovoltaic spectroscopy) to investigate the surface charge after light stimulation. As shown in Figure R3, the membrane surface becomes positively charged after the light irradiation. The responding positive voltage toward wavelength is well consistent with the UV-vis spectra.

Figure R3. SPV spectra of the porphyrin channel membrane.

To verify the ion selectivity difference, we have also conducted the I-V curves under a 50-fold concentration gradient to quantitatively evaluate the anion selectivity. As shown in Figure R4,

the diffusion potential of the porphyrin channel membrane increases from 85.4 mV to 123.5 mV when the light irradiation is on the high-concentration side, indicating the increment of anion selectivity.

Figure R4. The ion selectivity transformation toward light stimulation. (a) Current-voltage curves of the *p*-BCP@4TPP membrane in a 50-fold concentration gradient with and without light irradiation. (b) The changes in the diffusion potential and diffusion current responding to light stimulation.

We have added the relevant results and descriptions to the revised manuscript and revised supplementary information.

Comment 3. Figure 2g indicates that the developed membrane can output a very high power density of 17.4 W/m² under a 50-fold KCl gradient, compared with the previously reported values of 4-7 W/M² (e.g., Nano Energy 2023, 105, 108007; Adv. Funct. Mater. 2023, 33, 2211316; Adv. Funct. Mater. 2022, 32, 2204068). Please explain why so high performance compared with existing works.

Reply: Thanks for your valuable comments.

The high efficiency of osmotic energy conversion contributes to three aspects: (1) The high-density helical porphyrin channel. Despite the ultralow porphyrin content, the porphyrin aggregates into a high-density porphyrin channel array, providing the ion transport way. (2) The short transport distance between adjacent porphyrins. Chloride selectively transports from one porphyrin to the next porphyrin. The short *d*-spacing represents a short transport distance, meaning a low transport resistance. (3) Suppressed water swelling. The ultralow ion exchange capacity indicates low swelling, greatly suppressing the permselectivity loss

We have added relevant descriptions to the revised manuscript and cited the related literatures. “Compared to previous work, the ultrahigh power density of helical porphyrin membrane contributes to three aspects: the high-density helical porphyrin channel, the continuous and short ion transport pathway, as well as swelling suppression.”

Comment 4. What is the testing area for measuring ion transport and osmotic power?

Reply: Thanks for your valuable comments.

The testing area for electrochemical measurements is $8000 \mu\text{m}^2$. We have added this experimental detail to the revised supplementary information.

Comment 5. Figure 2e shows high Cl^- anion selectivity of the proposed membrane, compared with other anions, which is one of the main contributions of this work. I am curious how to measure in situ measure the current change when replacing the electrolyte. The detailed experimental measurement method should be mentioned. Another similar study of highly anion selective membrane (Sci. Adv. 2021, 7, eabe9924) should be credited.

Reply: Thanks for your valuable comments and helpful suggestions.

The ionic current is measured with different electrolytes indocently. To provide a comparison, we put these data into a figure in the order of the current magnitude. We are sorry for causing the misleading. We have provided the separate I-T curves for different electrolytes in Figure R5.

Figure R5. Current-time curves of the p-BCP@4TPP membrane with different electrolytes, showing the highest ion transport current for Cl^- .

We have updated Figure 2d. Thanks again for your kind reminder.

Just as you mentioned, the determination of chloride selectivity is indeed inspired by the previous work (Sci. Adv. 2021, 7, eabe9924), which has been cited in the revised manuscript.

Comment 6. I am curious why a fixed 2-fold or 3.25-fold concentration gradient is applied in Figure 5? Any physical meaning?

Reply: Thanks for your valuable comments.

The different concentration gradients are applied to determine the photo-induced driving force. Thus, we have applied various concentration gradients to test the effect, including a 2.0-fold concentration gradient. In this condition, Cl^- could selectively diffuse from the low-concentration side to the high-concentration side. To determine the kinetic process of the light-driven ion diffusion, we increase the concentration gradient from 2.0-fold to 3.25-fold. As the electrochemical results show, when the concentration gradient increases to 3.25-fold, the driving force based on the light-induced electric field could not overcome the resistance of concentration differences, achieving the active Cl^- transport (Figure R6).

Figure R6. The time-dependent photocurrent of the *p*-BCP@4TPP membrane under a (a) 2.0-fold, (b) 2.25-fold, (c) 2.5-fold, (d) 2.75-fold, (e) 3.0-fold, and (f) 3.25-fold concentration gradient with different light densities. The *p*-BCP@4TPP membrane achieves Cl^- pump properties under 2.0-fold, 2.25-fold, 2.5-fold, 2.75-fold, and 3.0-fold concentration gradients.

To provide a better understanding, we have added relevant descriptions to the revised manuscript. “Even under a 3-fold gradient, the built-in electric field can achieve Cl⁻ pump (Supplementary Figs. 34 and 35). When the gradient reaches 3.25-fold, the force of the concentration gradient dominates the ion transport, as Cl⁻ migrates from the high-concentration side to the low-concentration side (Figs. 5d and 5e).”

Comment 7. What is the structural stability (e.g., SEM, TEM, and x-ray data) of the proposed membrane after light stimulation for a while (e.g., half or one hour)?

Reply: Thanks for your valuable comments.

Following your suggestion, we have conducted the TEM, GI-SAXS, and WAXD tests to confirm the structural stability of the porphyrin channel membrane.

At the nanometer scale, The TEM image indicates that the membrane remains the nanocylinder structure after light stimulation for 1 hour (Figure R7a). We also conducted GI-SAXS tests to investigate the stability of periodic characteristics and alignment. As shown in Figure R7b, the scattering holes are observed in the equator direction on the 2D GI-SAXS pattern, showing the perpendicular alignment of the periodic nanocylinder. The perpendicular nanocylinder structure of the porphyrin channel membrane maintains excellent stability toward light stimulation.

Figure R7. The nanostructure stability of porphyrin channel membrane toward light stimulation. (a) TEM image of the porphyrin channel membrane after light irradiation for 1 hour. (b) 2D GI-SAXS patterns of the porphyrin channel membrane after light irradiation for 1 hour.

At the sub-nanometer scale, the WAXD test was carried out to study the π - π stacking of porphyrins. From the 2D pattern and 1D curves, the diffraction peak for porphyrin π - π stacking

remains almost unchanged since the porphyrin channel membrane is irradiated for 1 hour (Figure R8). The porphyrin aggregation structure shows excellent stability toward light simulation.

Figure R8. The porphyrin aggregate stability toward light stimulation. (a) 2D WAXD pattern of the porphyrin channel membrane after light irradiation for 1 hour. (b) The WAXD curves of the porphyrin channel membrane before and after light irradiation.

We have added these figures and relevant descriptions to the revised manuscript and revised supplementary information. “The porphyrin channel membrane shows structural stability toward light irradiation as the nanocylinder nanostructure and the porphyrin aggregation remains almost unchanged after the light stimulation (Supplementary Fig. 24 and Fig. 25).”

Comment 8. Any data to support the π - π stacking?

Reply: Thanks for your valuable comments.

We have taken the 2D WAXD test to confirm the π - π stacking of porphyrins. As shown in Figure R9, the scattering rings (highlighted with red color line) corresponding to porphyrin π - π stacking are seen from all samples.

Figure R9. 2D WAXD patterns of BCP membranes with different contents of TPP doping. The scattering ring corresponding to the porphyrin π - π stacking is seen from all samples.

We have updated these data in the revised supplementary information.

Reviewer #2 (Remarks to the Author):

This manuscript reports the development of a bioinspired light-driven chloride pump. The membrane pump is made using star block blocks with porphyrins at their center. Additional porphyrins are doped into the membrane to help create continuous channels across the membrane. A variety of characterization and transport tests are utilized to demonstrate the structure and interesting transport characteristics of the resulting films. The manuscript may be of interest to the readership of your journal but significant work is needed to communicate the main findings of the manuscript in a clearer and more concise manner. The authors have clearly worked hard to develop a detailed study of the light-driven chloride pumps. However, the manuscript presents the results of this extensive investigation without much detailed analysis or communicating clearly the hypothesis or motivation behind each experiment. I strongly suggest that the authors consider what information is absolutely essential for communicating their main findings and reduce the complexity of the figures. I think this will make assessing the underlying technological advances easier.

Reply: Thanks for your positive comment and valuable suggestions. We have reorganized Figure 2 and Figure 4 to strengthen the theme of the manuscript, as shown in Figures R1 and R2. We have also rewritten the ion pump part to provide a better understanding.

Figure R1. The specific Cl⁻ selectivity with doping-repaired helical porphyrin channels.

Figure R2. Photoelectric energy conversion with doping-repaired helical porphyrin channels.

Comment 1: A small detail that is important. The authors reference the porphyrin content of the copolymer materials. However, within the main text, I don't believe they ever clarify if it is a mole, mass, or volume percent.

Reply: Thanks for your valuable comment.

We have revised the description in the revised manuscript as “the mass content of porphyrin is less than 1%”.

Comment 2: It is important for the authors to execute control experiments based around star copolymer membranes that do not contain the porphyrin.

Reply: Thanks for your helpful suggestion.

Porphyrin in the block copolymer functions ion transport sites. Without porphyrins, the ion can only be non-selectively transported between gaps in the polymer chains, showing huge transport resistance and ultralow transport efficiency. Only the anion-selective porphyrin aggregates, efficient ion transport channels form, showing high ion conductivity.

To confirm the importance of porphyrin in ion transport, we have prepared a block copolymer membrane (PMMA-*b*-PS) without porphyrins. Without porphyrin, the block copolymer shows a white color, while the sample with porphyrin shows a red color (Figure R3a and R3b). Since there exist ion transport sites in the PMMA-*b*-PS membrane, the ions almost can not transport

through the membrane, performing a negligible ion conductivity from the I-V curve (Figure R3c).

Figure R3. (a) Photograph of PMMA-PS without porphyrin, showing white color. (b) Photograph of PMMA-PS with porphyrin, showing red color. (c) I-V curves of PMMA-*b*-PS BCP membrane and porphyrin-core BCP membrane, exhibiting huge transport resistance for the membrane without porphyrin.

Comment 3: The reported membranes are selective toward the transport of chloride. This appears to be true whether the membrane surface is illuminated or not. Does the selectivity of the membrane change at all when the surface is illuminated. The authors should consider executing experiments with a reverse bias. If the chloride was attempting to permeate through the negatively charged unilluminated side, transport should be slow due to electrostatic repulsion and an asymmetric I-V curve distinct from those shown in Figure S15 should arise. This experimental result would also help to support the authors' hypothesis regarding charge redistribution driving transport upon illumination.

Reply: Thanks for your helpful suggestion.

We have applied the I-V tests under an extremely asymmetric concentration gradient (0.5 M / 10 μ M) to investigate the ion selectivity. Since the concentration gradient is up to 5×10^4 . The ionic current is mainly contributed by the ion diffusion from the high-concentration side to the low-concentration side. Thus, the current at positive bias indicates the efficiency of Cl^- diffusion. The current at negative bias represents the efficiency of K^+ diffusion. Since the current at positive bias is much higher than the current at negative bias, the porphyrin channel membrane preferentially transports Cl^- . When the light stimulation is placed on the low-concentration side, the light-induced electric field impedes Cl^- transport, resulting in a lower current at positive

bias. When the light is irradiated on the high-concentration side, the light-induced electric field fastens Cl^- migration, achieving a higher current at a positive bias (Figure R4).

Figure R4. The ion selectivity of the membrane toward light irradiation. (a) The illustrated diagram of the ion transport without and with light irradiation. (b) I-V curves of the membrane under an extremely asymmetric concentration gradient.

We have added the results and descriptions to the revised manuscript and revised supplementary information. “It is worth noting that the Cl^- selectivity is enhanced upon light radiation (Supplementary Fig. 23).”

Comment 4: The authors should discuss the energetic efficiency of the light-driven transport process.

Reply: Thanks for your helpful suggestion.

Under a symmetric solution, the light-induced Cl^- migration achieves $56.0 \text{ mW} \cdot \text{m}^{-2}$ output power density (Fig. 4c). In this case, the membrane resistance is equal to the external resistance. The produced electric power of the whole system is twice the output power. The energy conversion efficiency (η) between light and electricity is calculated to be: $\eta = 2 \times P_1 / P_2$, where P_1 and P_2 represent the outpower density of the membrane ($56.0 \text{ mW} \cdot \text{m}^{-2}$) and light power density on the membrane ($4.2 \text{ mW} \cdot \text{cm}^{-2} = 42000 \text{ mW} \cdot \text{m}^{-2}$). The value is calculated to be $\sim 0.27\%$. We have added the above discussions to the revised manuscript and revised supplementary information.

Comment 5: I am not convinced that the numerical simulations provide useful insights since the results are controlled by the boundary conditions, which are set by the authors. The

comparison that is made is really just manifesting change that was a direct result of the researchers' input not a phenomenon that emerges naturally from the interactions within the system.

Reply: Thanks for your valuable comment.

The simulation results are indeed determined by the boundary conditions. The purpose of PNP simulation is to confirm that the redistributed surface charge could influence the ion transport property. To avoid misleading, we removed the simulation results for different concentration gradients. We have reconducted the simulation to just investigate the effect of variable surface charge density on the ion transport direction. The channel length is determined by the cross-sectional SEM image. We set five surface charge densities as: ① $\varepsilon=0.01 \text{ C}\cdot\text{m}^{-2}-0.02 \text{ C}\cdot\text{m}^{-2}\times X/L$, ② $\varepsilon=0.03 \text{ C}\cdot\text{m}^{-2}-0.06 \text{ C}\cdot\text{m}^{-2}\times X/L$, ③ $\varepsilon=0.06 \text{ C}\cdot\text{m}^{-2}-0.12 \text{ C}\cdot\text{m}^{-2}\times X/L$, ④ $\varepsilon=0.09 \text{ C}\cdot\text{m}^{-2}-0.18 \text{ C}\cdot\text{m}^{-2}\times X/L$, and ⑤ $\varepsilon=0.12 \text{ C}\cdot\text{m}^{-2}-0.24 \text{ C}\cdot\text{m}^{-2}\times X/L$, where X and L represent the X -axial position along the nanochannel and the nanochannel length. When the redistributed electric field is strong enough, the reverse Cl^- concentration along the porphyrin channel could happen (Figure R5). This is also the reason that the porphyrin channel membrane could only achieve active Cl^- migration under a 3.0-fold concentration gradient with a light density of $100.8 \text{ mW}\cdot\text{cm}^{-2}$.

Figure R5. (a) The illustrated diagram of redistributed surface charge density for the porphyrin channel. The calculated Cl^- concentration distribution profiles along the X-axis of a single porphyrin channel under different surface charge densities (b-f). The inversed Cl^- concentration distribution indicates the inversed ion migration direction with light irradiation.

Based on the above results, we have revised the descriptions of the simulations. “The light-driven ion pump property is also semi-quantitatively confirmed by simulation results. The light irradiation redistributes the surface charge density of the membrane, resulting in the illustrated side being positively charged and the unillustrated side being negatively charged. As the difference in charge density between the two sides increases, the concentration gradient along the porphyrin channel would reverse. The ion accumulation at the low-concentration side and the ion depletion at the high-concentration side could drive the Cl⁻ migration against the concentration gradient difference.”

Comment 6: The concentration profile in Figure 5f deserve further analysis and discussion. Why don't the concentrations at the boundary of the two cases align with each other? Additionally, the local curvature of the concentration profiles for the illuminated systems suggest that transport is being driven from right to left (similar to the dark cases). More details regarding these simulations are needed for other to be able to repeat them.

Reply: Thanks for your helpful suggestion.

Generally, the channel with the higher surface charge density attracts more Cl⁻. The Cl⁻ concentration would gradually decrease from the illustrated side to the unillustrated side. But boundary Cl⁻ concentrations on the entrance of the porphyrin channel are influenced by the bulk solution. The low Cl⁻ concentration in bulk solution weakens the ion accumulation at the illustrated entrance. Meanwhile, the high Cl⁻ concentration in bulk solution weakens the ion depletion at the unillustrated entrance. We have also added the simulation models to the revised supplementary information (Figure R6).

Figure R6. Calculation of porphyrin channel model (not in scale). The theoretical simulation is based on the coupled two-dimensional Poisson-Nernst-Planck equations.

We have added the above descriptions to the revised manuscript. “The concentration decrement at the low-concentration entrance and the concentration increment at the high-concentration entrance are contributed from the bulk solution”.

Reviewer #3 (Remarks to the Author):

This manuscript reports a photoactive helical porphyrin channel that shows specific chloride selectivity. It is demonstrated that this porphyrin channel can act like a chloride ion pump under light irradiation, which drives Cl^- migration against a 3-fold concentration gradient. Although a series of experiments have been conducted in this work, there are problems with the experimental results and explanations. Therefore, we do not think this work can be published at the present stage.

Reply: Thanks for your helpful comments. And we will respond to your comment point-in-point as follows.

Comment 1. It is not necessary to emphasize “doping-repaired” in the title. The core of this work is the porphyrin-channel based chloride ion pump. The whole system can also work with undoped helical porphyrin channel.

Reply: Thanks for your constructive suggestions.

As you said, the undoped porphyrin channels do have a photoactive effect. Following your constructive suggestion, we have revised the title to “Bioinspired light-driven chloride pump with helical porphyrin channels”.

Comment 2. It is known that porphyrin molecules are generally negatively charged and easily coordinate with metal ions. Therefore, it is confusing that the porphyrin channel preferentially binds to anions and shows anion selectivity in this work.

Reply: Thanks for your helpful comments.

As shown in Figure R1, the porphyrin can form dynamic porphyrin–chloride complexes with Cl^- ions through hydrogen bonding (J. Phys. Chem. A, 2005, 109, 7442). It is easy to attract and diffuse Cl^- ions through porphyrin channels.

But for monovalent metal ion, like K^+ , Na^+ , the coordination is too weak. It is hard for porphyrin to adsorb monovalent metal ions on the surface. While, for divalent and trivalent metal ions such as Cu^{2+} , Fe^{2+} and Fe^{3+} can be incorporated into the porphyrin cavity, leading to very stable inner complexes. Thus, they are difficult to transport once the complex is formed. For

monovalent metal ion, the adsorption is the rate determining step. For divalent and trivalent metal ions, the diffusion is the rate determining step. Therefore, from the interaction analysis, the porphyrin channels have Cl^- selectivity over cations.

Figure R1. The molecular structure of the hydrogen bond. (a) The hydrogen bond between Cl^- and porphyrin (partial structure). (b) Schematic graph of porphyrin species interacting with chloride anion. The green atoms represent chloride anion.

Comment 3. What is the isoelectric point of a porphyrin molecule? The zeta potential of the porphyrin-core star block copolymer needs to be measured.

Reply: Thanks for your constructive suggestion.

Following your valuable suggestion, we have conducted the zeta potential test of the *p*-BCP@4PP membrane. As shown in Figure R2, the membrane is positively charged under acidic conditions, and the isoelectric point of a porphyrin molecule is about 4.7. As the zeta potential of the membrane was tested with 1 mM KCl solution, the porphyrin attracts Cl^- above 4.7, showing a negative zeta potential.

Figure R2. Zeta potential of the porphyrin membrane, showing positive membrane surface under acidic conditions.

We have added the results to the supplementary information.

Comment 4. The influence of pH on the chloride selectivity and chloride ion pump should be given. It is a very important control experiment.

Reply: Thanks for your helpful comments.

Following your constructive suggestions, we first conducted the I-V curves under a 50-fold concentration gradient using different pH electrolytes. As shown in Figure R3, the porphyrin becomes protonated under the acidic condition, increasing the surface charge density. Thus, the diffusion potential experiences an increment from alkaline to acidic electrolyte. The diffusion potential is 56.9 mV, 64.4 mV, and 73.6 mV corresponding to the pH value of 11.0, 7.0, and 3.0, respectively. Based on the Goldman-Hodgkin-Katz voltage equation, the ion selectivity could be calculated to be 12.2, 18.6, and 34.0, respectively.

Figure R3. The Cl^- selectivity toward pH value. (a) The illustrated diagram of the protonated porphyrin under acidic conditions. (b) The I-V curves of the membrane under a 50-fold concentration (0.5 M / 10 mM) using different pH electrolytes.

We further investigated the effect of pH value on the ion pump property. Under acidic conditions, the light-induced electric field could drive Cl^- migration against a 2.0-fold concentration gradient (Figure R4). While under alkalic conditions, the membrane could achieve active Cl^- migration against a 4.0-fold concentration gradient with light irradiation (Figure R5). Under neutral conditions, the porphyrin channel membrane achieves Cl^- migration against a 3.0-fold concentration gradient (Figure 5f). The differences in ion pump properties arise from the differences in surface charge densities. The higher surface charge density (acidic

condition) indicates the weaker redistributed surface charge density with light irradiation, resulting in a smaller light-driven force.

Figure R4. The time-dependent photocurrent cycle of the *p*-BCP@4TPP membrane under (a) 10 mM / 20 mM and (b) 10 mM/ 30 mM concentration gradient (pH=3.0) with a light density of $100.8 \text{ mW}\cdot\text{cm}^{-2}$.

Figure R5. The time-dependent photocurrent cycle of the *p*-BCP@4TPP membrane under (a) 10 mM / 20 mM, (b) 10 mM/ 30 mM, (c) 10 mM/ 40 mM, and (d) 10 mM/ 50 mM concentration gradient (pH=11.0) with a light density of $100.8 \text{ mW}\cdot\text{cm}^{-2}$.

We have added these Figures and descriptions to the revised manuscript and revised supplementary information.

Comment 5. In Figure 1c, a clearer and enlarged TEM image is required. What is the thickness of the membrane? The SEM image of the cross section of the membrane should be given.

Reply: Thanks for your helpful suggestions.

Following your helpful suggestion, we have carried out a clearer and enlarged TEM image and revised Figure 1c, as shown in Figure R6. We have updated Figure 1c.

Figure R6. TEM image of BCP self-assembled membrane, exhibiting periodic nanocylinder structure. Scale bar: 500 nm. Inset: enlarged TEM image with a higher resolution.

As you suggested, we have performed cross-section SEM test. As shown in Figure R7, we can see the transmembrane nanocylinder structure. The membrane thickness is about ~ 120 nm.

Based on the membrane thickness, we have updated the simulation results. We have added the SEM image and relevant descriptions to the revised manuscript and revised supplementary information.

“From the cross-section SEM image (Supplementary Fig. 7), we can see transmembrane nanocylinder structure.”

Figure R7. The SEM image of the membrane cross-section, indicating the transmembrane nanocylinder structure (with an average thickness of ~120 nm).

Comment 6. The salt bridge is not shown in Figure S17. The photo of the actual device used in this work also needs to be shown.

Reply: Thanks for your helpful suggestions.

We have revised Figure S17, as shown in Figure R8.

Figure R8. Illustrated diagram of the light-driven ion transport test.

Comment 7. About the anion/cation selectivity of the membrane, a more intuitive and effective experiment is recommended, which can be referred to “Figure 4a” in a previous work (DOI: 10.1021/jacs.5b09918).

Reply: Thanks for your constructive suggestions.

Following your helpful suggestions, we have performed the I-V curves of the membrane under an ultrahigh asymmetric concentration gradient (0.5 M / 10 μ M). The ionic currents are mainly contributed by the ion diffusion from the high-concentration side to the low-concentration side. The ionic current at the positive voltage (Cl^-) is prominently higher than the ionic current (K^+) at the negative voltage, indicating much higher Cl^- transport efficiency (Figure R9).

Figure R9. I-V curve of the *p*-BCP@TPP membrane under an ultrahigh asymmetric concentration gradient (0.5 M / 10 μ M).

We have added the data and relevant description to the revised manuscript and revised supplementary information. “The Cl^- selectivity is experimentally researched by the I-V curve under an extremely asymmetric concentration gradient (0.5 M / 10 μ M). The ion migration contribution to the ionic current from the low-concentration side is negligible. As shown in Supplementary Fig. 16, the ionic current at the positive voltage (Cl^-) is prominently higher than the ionic current at the negative voltage (K^+), indicating much higher Cl^- transport efficiency.”

Comment 8. “Page 5 Line 136” It is described that “the Cl^- selectivity is contributed by the formation of halogen bonds”. What is the molecular structure of this “halogen bond”? In addition, ref.33 is quoted here to explain this selectivity. However, the porphyrin molecules reported in Ref.33 are metal-ion centered, which is a totally different story. An explanation is needed here.

Reply: Thanks for your kind reminder and helpful comment.

We have revised the “halogen bond” as the “hydrogen bond”.

As shown in Figure R10, the porphyrin can form dynamic porphyrin–chloride complexes with Cl^- ions through hydrogen bonding (J. Phys. Chem. A, 2005, 109, 7442). It is easy to attract and diffuse Cl^- ions through porphyrin channels.

But for monovalent metal ion, like K^+ , Na^+ , the coordination is too weak. It is hard for porphyrin to adsorb monovalent metal ions on the surface. While, for divalent and trivalent metal ions such as Cu^{2+} , Fe^{2+} and Fe^{3+} can be incorporated into the porphyrin cavity, leading to very stable inner complexes. Thus, they are difficult to transport once the complex is formed. For monovalent metal ion, the adsorption is the rate determining step. For divalent and trivalent metal ions, the diffusion is the rate determining step. Therefore, from the interaction analysis, the porphyrin channels have Cl^- selectivity over cations.

Figure R10. The molecular structure of the hydrogen bond. (a) The hydrogen bond between Cl^- and porphyrin (partial structure). (b) Schematic graph of porphyrin species interacting with chloride anion. The green atoms represent chloride anion.

We have added the illustrated diagram to the revised supplementary information.

Comment 9. The light irradiation might also cause the increase of membrane temperature or even a temperature gradient inside the membrane. This will not only increase the ion transport rate, but also drive the ion transport along the temperature gradient. Therefore, the influence of temperature should also be discussed.

Reply: Thanks for your constructive suggestions.

Increasing the membrane temperature indeed enhances ion transport, as shown by the I-V curves under different temperatures (Figure R11). To investigate the light-induced surface temperature increase, we have taken infrared imaging tests. As shown in Figure R11, the membrane temperature experiences a very slight increment (less than $1\text{ }^\circ\text{C}$) after light

irradiation (Figure R12). Therefore, the ion transporting enhancement contributed by UV-induced temperature increase can be negligible. There are three factors contributing to the negligible temperature changes. First, the content of porphyrin is extremely low. The thermal effect is not obvious. Second, we applied a weak light source, whose light density is comparable to the sunlight. Third, the membrane is immersed in water, the tiny heat is conducted quickly.

Figure R11. The ionic transport property of the *p*-BCP@4TPP membrane toward different temperatures. (a) I-V curves of the *p*-BCP@4TPP membrane toward different temperatures. (b) The ionic conductivity of the *p*-BCP@4TPP membrane toward different temperatures, showing the higher ionic conductivity for the higher temperatures.

Figure R12. Infrared images of the *p*-BCP@4TPP membrane under different light densities. (a) Photograph of the *p*-BCP@4TPP membrane (Scale bar: 1 cm). The infrared image of the *p*-BCP@4TPP membrane (b) without light, with (c) 60.0 mW · cm⁻², (d) 80.4 mW · cm⁻², (e) 90.6

$\text{mW} \cdot \text{cm}^{-2}$, and (f) $100.8 \text{ mW} \cdot \text{cm}^{-2}$ (Scale bar: 0.5 cm).

Following your helpful suggestions, we have added the infrared imaging results and relevant descriptions to the revised manuscript and revised supplementary information. “The light induced positive charge on surface is confirmed by the surface photovoltage spectroscopy test (Supplementary Fig. 28). Although the light irradiation might also cause the increase of membrane temperature, the contribution can be negligible (Supplementary Fig. 29), because the content of porphyrins is low, while the light is weak, and meanwhile, the tiny heat is quickly conducted by the electrolyte.”

Comment 10. It is described that the ion pump is mainly caused by the light-induced surface charge redistribution. However, when the surface charge is redistributed, the whole system should reach a steady state without any further ion migration. What causes the continuous ion flow?

Reply: Thanks for your helpful comment.

Just as you said, the light irradiation induces the surface charge redistribution and produces the built-in electric field. Under a specific light density, the electric field reaches a steady value, maintaining a driven force. When the light-induced electric force is larger than the force from the concentration gradient. The ion migration against the concentration gradient difference happens. As long as the light stimulation exists, the driving force of the electric field can always exist, causing continuous ion migration. When the light stimulation is removed, the ions would migrate along the direction of the concentration gradient difference.

Comment 11. The redistribution of the surface charge under light irradiation may also lead to a reversal of ion selectivity. This ion selectivity reversion can also cause the reversion of the direction of current and membrane potential, showing a similar data reported in this work. The anion/cation selectivity of the membrane under light irradiation needs to be measured. Further discussion needs to be added in the manuscript.

Reply: Thanks for your constructive suggestions.

To investigate the ion selectivity, we have performed I-V curves of the membrane under an

extremely asymmetric concentration gradient (0.5 M / 10 μ M). Without light irradiation, the membrane shows the Cl⁻ selectivity (Figure R13, black line). When the light stimulation is placed on the low-concentration side, the ionic current at the positive voltage (Cl⁻) experiences a slight decrement, indicating that the light-induced electric field impedes the Cl⁻ diffusion (Figure R13, red line). When the light simulation is placed on the high-concentration side, the ionic current at the positive voltage (Cl⁻) exhibits an obvious increment, indicating that the light irradiation fastens the Cl⁻ migration (Figure R13, blue line).

Figure R13. The ion selectivity of the membrane toward light irradiation. (a) The illustrated diagram of the ion transport without and with light irradiation. (b) I-V curves of the membrane under an extremely asymmetric concentration gradient.

We have added the figures and relevant descriptions to the revised manuscript and revised supplementary information. “The light-increased ionic current arises from the light-increased Cl⁻ selectivity”

Comment 12. Scale bar of current in Figure 3e and Figure 4b,d,g is missing.

Reply: Thanks for your constructive suggestion.

We have added the scale bar to these figures, which are shown in Figure R14.

Figure R14. Revised I-T curves with the scale bars.

REVIEWER COMMENTS

Reviewer #1 (Remarks to the Author):

The manuscript has been well revised in light of my previous comments. Now there is only one minor typo which should be corrected before publication; that is, the journal in Ref. 37 is Nano Energy.

Reviewer #2 (Remarks to the Author):

The authors have done a nice job of addressing the concerns raised by the reviewers. I believe that this manuscript would be of interest to the readership of this journal.

Reviewer #3 (Remarks to the Author):

The authors have responded to most of the raised comments. However, some cases need to be further explained.

1. "Reviewer 3 Comment 2" It is said that "For monovalent metal ion, the adsorption is the rate determining step. For divalent and trivalent metal ions, the diffusion is the rate determining step." Is there any evidence to support this?
2. "Reviewer 3 Comment 3" First, if a membrane is negatively charged in 1 mM KCl solution, the membrane should be cation selective due to the electrostatic attraction. Secondly, if the membrane is negatively charged at pH 7 due to Cl⁻ adsorption, the membrane should be more negatively charged because there will be an additional electrostatic attraction. Thirdly, as far as we know, the porphyrin is negatively charged at pH 3 because of the deprotonation of the molecule.
3. "Reviewer 3 Comment 7" Why the ionic current at the positive voltage is assigned as the "Cl⁻" current? If the working electrode is put in the high-concentration side, the ionic current at the positive voltage could be "K⁺" current.

Reply to Reviews Comments

Reviewer #1 (Remarks to the Author):

Comment: The manuscript has been well revised in light of my previous comments. Now there is only one minor typo which should be corrected before publication; that is, the journal in Ref. 37 is Nano Energy.

Reply: Thanks for your helpful comments and constructive suggestions. We have corrected the citation in the revised manuscript.

Reviewer #2 (Remarks to the Author):

Comment: The authors have done a nice job of addressing the concerns raised by the reviewers. I believe that this manuscript would be of interest to the readership of this journal.

Reply: Thanks for your valuable comments.

Reviewer #3 (Remarks to the Author):

Comment: The authors have responded to most of the raised comments. However, some cases need to be further explained.

Reply: Thanks for your positive comments and constructive suggestions, which truly improve our work. We will respond to your concerns point-in-point as follows.

Comment 1. “Reviewer 3 Comment 2” It is said that “For monovalent metal ion, the adsorption is the rate determining step. For divalent and trivalent metal ions, the diffusion is the rate determining step.” Is there any evidence to support this?

Reply: Thanks for your valuable comments.

To verify the Cl selectivity toward cations, such as K^+ , Mg^{2+} , and Al^{3+} , we have conducted the simulation based on density functional theory (DFT) calculations. The cation dissociation process from porphyrin is simulated and the relative free energy barriers for different cations transport are shown in Figure R1, which are much higher than that of Cl^- transport ($0.6 \text{ kcal} \cdot \text{mol}^{-1}$, Supplementary Fig. 15). In addition, for cations (K^+ , Mg^{2+} , and Al^{3+}) with different valence, the dissociation energy barriers of cations increase with the increase of valence, indicating that the higher the valence of the cations, the stronger the interaction with porphyrin.

Figure R1. The relative free energy barriers of scan results for different cations (scanning the N-X (X= K^+ , Mg^{2+} or Al^{3+}) bond in red color), relative free energy barriers are in $\text{kcal} \cdot \text{mol}^{-1}$. For monovalent ions such as K^+ , the weak interaction between the ion and porphyrin does not

favor the K^+ enrichment on the porphyrin channel surface. As shown in Figure R2, the Cl^- concentration on the porphyrin aggregates is much higher, showing a much stronger affinity between Cl^- and porphyrin.

Figure R2. The strong affinity between porphyrin and Cl^- . (a) TEM image of porphyrin aggregates. (b) Energy dispersive X-ray (EDX) mapping of Cl^- with the porphyrin aggregates. (c) EDX mapping of K^+ with the porphyrin aggregates.

We have added these data to the revised manuscript and revised supplementary information. We have also added the following discussions:

“Compared to K^+ , Cl^- shows stronger affinity with porphyrin. As a result, the porphyrin channel shows Cl^- selectivity towards K^+ (Supplementary Fig. 16)”.

“The Cl^- selectivity is also confirmed by the simulation results, as the relative free energy barrier for Cl^- migration is much lower than those of cations (Supplementary Fig. 18).”

Comment 2. “Reviewer 3 Comment 3” First, if a membrane is negatively charged in 1 mM KCl solution, the membrane should be cation selective due to the electrostatic attraction. Secondly, if the membrane is negatively charged at pH 7 due to Cl^- adsorption, the membrane should be more negatively charged because there will be an additional electrostatic attraction. Thirdly, as far as we know, the porphyrin is negatively charged at pH 3 because of the deprotonation of the molecule.

Reply: Thanks for your valuable comments.

The Porphyrin core has three forms (shown in Figure R3): free-base porphyrin (uncharged), protonated porphyrin (positively charged), and deprotonated porphyrin (negatively charged). Since the N-H groups of the porphyrin core are very weakly acidic ($pH > 15$), the porphyrin

channel could not be deprotonated in the testing solutions (*J. Phys. Chem. B* 2006, 110, 587; *J. Phys. Chem. B* 2004, 108, 10185).

The pKa for porphyrin protonation is influenced by the macrocycle substitution. The pKa value is generally ranging from 3.0 to 7.0 (*J. Phys. Chem. C* 2014, 118, 9196; *Biomacromolecules* 2012, 13, 60; *Anal. Chim. Acta* 2011, 705, 306). From the surface zeta potential of the porphyrin, the pKa of the membrane is ~ 4.6 . Below this pH value, the porphyrin is protonated with H^+ , showing positive potential. Above this pH value, the porphyrin is free-base, but the Cl^- could be absorbed by the membrane surface by hydrogen bond, showing a negative membrane surface charge. The zeta potential of the porphyrin channel membrane toward variable pH values is similar to the anion-selective P4VP channels ($pK_a \approx 5.2$ for P4VP) (*J. Membr. Sci.* 2016, 501, 161; *ACS Appl. Mater. & Interfaces* 2020, 12, 55116; *J. Am. Chem. Soc.* 2015, 137, 14765). For free-base porphyrin in neutral conditions, because of stronger affinity with Cl^- and lower Cl^- migration resistance than K^+ , the porphyrin channel exhibits excellent Cl^- selectivity.

Figure R3. Three forms of porphyrin: free-base porphyrin, protonated porphyrin, and deprotonated porphyrin.

Comment 3. “Reviewer 3 Comment 7” Why the ionic current at the positive voltage is assigned as the “ Cl^- ” current? If the working electrode is put in the high-concentration side, the ionic current at the positive voltage could be “ K^+ ” current.

Reply: Thanks for your valuable comments.

The electrochemical testing condition for the I-V curve under an ultra-asymmetric concentration gradient is depicted as follows: The positive electrode is placed on the low-concentration side (LC, 5 μM), while the negative electrode is placed on the high-concentration

side (HC, 0.5 M, see Figure R4a-I). Under positive bias (Figure R4a-II), the electric field direction is from the LC side to the HC side. Because the K^+ migration from the LC side to the HC side can be neglected, the current is mainly contributed by the Cl^- migration from the HC side to the LC side (Figure R4b, Cl^-).

Under negative bias (Figure R4a-III), the electric field direction is from the HC side to the LC side. Because the Cl^- migration from the LC side to the HC side can be neglected, the current is mainly contributed by the K^+ migration from the HC side to the LC side (Figure R4b, K^+).

The current value under positive bias is bigger than negative bias. It means the membrane shows Cl^- selectivity.

Figure R4. The Cl^- selectivity of the porphyrin channel membrane. (a) The electrochemical testing condition for the I-V curve under an ultra-asymmetric concentration gradient. The positive electrode is placed on the 0.5 M concentration gradient. (b) I-V curve of the *p*-BCP@4TPP membrane under an ultra-asymmetric concentration gradient (0.5 M / 5 μM).

We have revised the figure and added relevant descriptions in the revised supplementary information.

REVIEWERS' COMMENTS

Reviewer #3 (Remarks to the Author):

This revised manuscript can be accepted for publication.